# Abdominal Aortic Aneurysm Diameter versus Volume: A Systematic Review

**DOI:** 10.3390/biomedicines11030941

**Published:** 2023-03-17

**Authors:** Gediminas Vaitėnas, Valerija Mosenko, Austėja Račytė, Karolis Medelis, Arminas Skrebūnas, Tomas Baltrūnas

**Affiliations:** 1Faculty of Medicine, Vilnius University, 01513 Vilnius, Lithuania; valerijamosenko@gmail.com (V.M.); austejarace@gmail.com (A.R.); arminas.skrebunas@gmail.com (A.S.); tomas.baltrunas@gmail.com (T.B.); 2Center of Vascular and Endovascular Surgery, Vilnius University Hospital Santaros Klinikos, 08410 Vilnius, Lithuania; karolis.medelis@santa.lt

**Keywords:** abdominal aortic aneurysm, diameter, volume, follow-up

## Abstract

Recently, AAA volume measurement has been proposed as a potentially valuable surveillance method in situations when diameter measurement might fail. Objective: The aim of this systematic review was to analyze the results of previous studies comparing AAA diameter and volume measurements. Methods: A systematic search in PubMed, Cochrane, and EMBASE databases was performed to identify studies investigating the use of diameter and volume measurements in AAA diagnosis and prognosis in English, German, and Russian, published until December 2022. The manuscripts were reviewed by three researchers and scored on the quality of the research using MINORS criteria. Results: After screening 752 manuscripts, 19 studies (*n* = 1690) were included. The majority (*n* = 17) of the manuscripts appeared to favor volume. It is, however, important to highlight the heterogeneity of methodologies and lack of standardized protocol for measuring both volume and diameter in the included studies, which hindered the interpretation of the results. Conclusions: The clinical relevance of abdominal aortic aneurysm volume measurement is still unclear, although studies show favorable and promising results for volumetric changes in AAA, especially in follow-up after EVAR.

## 1. Introduction

To this day, aortic diameter is known to be a key parameter used not only for diagnosing abdominal aortic aneurysms (AAA) but also as a threshold for AAA elective repair and follow-up of already diagnosed aneurysms [1,2]. Although diameter measurements slightly depend on the imaging technology, whether it be ultrasound or computed tomography angiography (CTA), an abdominal aortic diameter larger than 3.0 cm or a diameter that is 1.5 times larger than normal is regarded as aneurysmatic [1]; 5.0 cm and 5.5 cm are considered to be threshold values for elective repair for women and men, respectively, whereas rapid diameter growth (>1 cm/year) requires timely referral to a vascular surgeon [1,3]. Nowadays, more than half of treatment procedures are performed endovascularly, and follow-up is recommended, indicating that even small changes in aortic size can be clinically significant.

Despite the worldwide use of diameter measurement for AAA diagnosis, surveillance, and clinical decision-making, there has been a debate about whether it is the most accurate and reliable method [4]. The accuracy of AAA diameter measurement might be distorted due to poor ability to detect shape changes, tortuosity of the aorta, and high rates of interobserver variability. Even though a larger AAA diameter is traditionally associated with a greater risk of aneurysm rupture, it is estimated that every year up to 2% of small AAAs rupture while some large diameter aneurysms remain stable along the course of a patient‘s life [5,6,7]. Some authors declare that diameter measurement is not able to detect small changes in aneurysm growth, thus, making this method not completely reliable in some clinical scenarios [8].

Recently, AAA volume measurement has been proposed as a potentially valuable surveillance method in situations when diameter measurement might fail. Abdominal aortic volume can be measured using several different techniques, such as three-dimensional reconstruction of computed tomography angiography (3D-CTA) or magnetic resonance angiography (3D-MRA) as well as three-dimensional ultrasound (3D-US), which is an emerging method in the field of AAA volume measurement [8,9]. Measurement of aortic volume is considered to be useful in defining the morphology of the aneurysmal sac in a three-dimensional way; also, it presumably has a higher value for surveillance after endovascular aneurysm repair (EVAR) [10]. Another advantage of AAA volume measurement could be the ability to accurately monitor saccular aneurysms because this type of aneurysm has a weaker relationship between the increased diameter and the risk of rupture [8]. Despite its benefits, volume assessment of AAA is not currently used in daily clinical practice. If performed manually, volume measurement requires skills and specialized software and is time-consuming, which makes it a less attractive method compared to diameter measurement. However, different automatic or semi-automatic segmentation software systems are currently under development in order to provide doctors with fast and accurate AAA volume measurements [11,12].

Notwithstanding the existing evidence that aortic volume measurement provides relevant information about the morphology of AAA, to this day, diameter measurement remains the gold standard for the assessment of this pathology. To the best of our knowledge, no systematic reviews comparing AAA diameter and volume measurements have been reported to date. The aim of this systematic review is to analyze the results of previous studies comparing AAA diameter and volume measurements.

## 2. Materials and Methods

### 2.1. Literature Search Strategy

The protocol for the planned systematic review was registered in the international register of systematic reviews (PROSPERO) [13]. Three authors performed an independent literature search to identify studies investigating the use of diameter and volume measurements in AAA diagnosis and prognosis. The PubMed database (https://pubmed.ncbi.nlm.nih.gov, accessed on 30 January 2023) was searched for papers published until 1 December 2022, using the following keywords: “Aortic Aneurysm, Abdominal” (MeSH) AND “Diameter” (MeSH) AND “Volume” (MeSH). Free text words were also used to avoid missing manuscripts that had not yet been given a MeSH label. A total of 370 studies were identified, and 38 studies were deemed eligible after reading the abstracts. The EMBASE database (https://www.embase.com/landing?status=grey, accessed on 30 January 2023) was checked for relevant studies as well, published until 1 December 2022, with the following keywords: “Aortic Aneurysm, Abdominal” (MeSH) AND “Diameter” (MeSH) AND “Volume” (MeSH). A total of 343 studies were found, and 11 were eligible. The Cochrane Database (https://www.cochranelibrary.com, accessed on 30 January 2023) of Systematic Reviews was searched until 1 December 2022 using the following words: “Abdominal aortic aneurysm”, “Volume”, and “Diameter”, with 39 reviews found, 0 eligible. Any disagreement the authors had was resolved after the independent reading of a full text. The literature search strategy, as well as article selection, is outlined in Figure 1, a flow chart of the systematic literature search according to PRISMA guidelines (PRISMA code: CRD4202339635) [14].

### 2.2. Inclusion and Exclusion Criteria

All studies included in this systematic review had to meet the selection criteria depicted in Table 1.

### 2.3. Types of Studies

Human studies.

### 2.4. Types of Participants

Patients with abdominal aortic aneurysm.

### 2.5. Types of Outcomes

The outcome measure was defined as the clinical relevance of diameter and volume measurement as well as the comparative usefulness of them in patients with AAA. To assess the relevance of AAA diameter and volume in clinical practice, we specifically reviewed manuscripts that correspond to the following parameters: adequate (>6) methodological index for non-randomized studies (MINORS) quality score, sensitivity, and specificity calculation. The results of sensitivity, specificity, positive predictive value, and negative predictive value calculations were included if provided in the manuscripts reviewed.

### 2.6. Data Extraction and Critical Appraisal

After identifying relevant titles, all abstracts were screened, and full-text manuscripts were accessed through Vilnius University VPN. A manual cross-reference search of references of included manuscripts was performed to identify other relevant studies. The validity assessment of each manuscript was performed using the MINORS quality score. Non-comparative studies were evaluated on eight quality items, while comparative studies were evaluated on twelve quality items. For each quality item, a score of 0 indicates that it was not reported in the manuscript, 1 indicates that it was reported inadequately, and 2 indicates that it was reported adequately. This adds up to a maximum MINORS score of 16 for non-comparative studies and 24 for comparative studies [15]. In this review, a score of ≤6 was considered poor quality. Quality assessment results of the studies included are demonstrated in Table 2.

## 3. Results

This systematic review analyzed 19 studies (*n* = 1690) comparing volume and diameter measurements in the diagnosis and prognosis of AAA. Two studies reported no prognostic difference between volume and diameter measurement, while seventeen manuscripts appeared to favor volume. The summary of studies and their outcomes are represented in Table 3 and Table 4, respectively. Studies including the endoleak data are marked with asterisks (*) in both tables.

## 4. Discussion

To this day, aortic diameter is considered to be a gold standard parameter used in clinical decision-making, while volume measurement is still in its infancy [1]. This systematic review, however, suggests that changes in aortic volume may outweigh the benefits of diameter measurement. On the other hand, it should be emphasized that none of the investigated studies researched the connection between changes in aneurysm volume and mortality. The majority of studies (17 of 19) found volume to be a better tool in characterizing AAAs, while the remaining 2 studies show no significant advantage in using volume measurements against diameter. It has to be noted that most studies favoring volume have a higher or at least equal MINORS score in comparison to studies that did not find a difference between volume and diameter measurements. It is important to highlight the heterogeneity of methodologies and lack of standardized protocol for measuring both volume and diameter in the included studies, which hindered the interpretation of the results.

Six studies included in this systematic review investigated endoleaks after EVAR. Five out of six studies found volume to be a more sensitive measurement in identifying endoleaks after EVAR than diameter. On the other hand, a study conducted by Quan et al. denies the superiority of volume measurement [19]. Quant et al. found that more endoleaks were present in the stable diameter patient group than in the stable volumetric patient group. Nevertheless, according to the authors, 10 patients developed endoleaks without volumetric change. Bargellini et al., on the contrary, found that a change in AAA volume of less than 0.3% at six months follow-up was the strongest independent predictor of endoleak [20]. Results presented by Bargellini et al. were not homogenous as well; a group of patients developed endoleaks without aortic enlargement, while some patients had increased aortic volumes that did not result in endoleaks.

Few of the studies compared the relation of volume and diameter measurements with peak wall stress (PWS) and peak wall rupture index (PWRI) [27,31]. Liljeqvist et al. found that volume growth correlated stronger with PWS and PWRI than diameter. Raghavan et al. found that volume had the strongest correlation with PWS. Such findings put volume measurement forward as being a potentially more feasible parameter for predicting the risk of rupture. The main concern is that this connection is not completely straightforward, as both PWS and PWRI are compound parameters.

Spanos et al. investigated the difference between ruptured and unruptured AAAs [6]. The mentioned study revealed that volumes were significantly different between the groups, while maximum diameter (Dmax) did not have a statistically significant difference. A threshold value of 380 mL was fairly well associated with rupture, accompanied by sensitivity and specificity of 60%. Dmax was not found to be a predictor of rupture.

Although this systematic review included studies comparing aortic diameter and volume measurement, several other potential prognostic markers regarding aneurysm growth and the risk of rupture can be mentioned. Some studies suggest that wall shear stress, wall thickness, and inflammatory markers might be valuable parameters to predict aneurysm rupture [32,33,34,35]. Additionally, hemodynamic features, such as tortuosity or the occlusion of aortic outflow, have been reported to be associated with an increased risk of rupture [36]. Nevertheless, these are complex parameters, and currently, their application in clinical practice is quite limited.

Although the majority of the studies analyzed in this systematic review place AAA volume superior to diameter, we believe that such results cannot be taken as an absolute. These findings might be partially influenced by the methodologies used to calculate diameter and volume. A number of different methods to calculate aortic diameter were used, for example, orthogonal and axial planes, antero–posterior, leading edge to leading edge, inner to inner, and outer to outer measurements [5,9,17,18,19,24,30]. It is important to highlight the lack of a standardized way to measure AAA volume. Most authors measure volume from below the lowest renal artery to the aortic bifurcation [5,6,16,19,20,22,24,25,28,30,31]. Others measure volume in the portion that includes the abdominal aorta, aneurysm, and iliac arteries covered by stent graft [17]. However, a significant portion of AAA volume measurement also involves iliac arteries, which could be measured likewise [21]. Mathematical methods of how centerline is calculated and how a specific target region of interest is chosen also may vary. Furthermore, different methods (CTA, US) are used to measure volume. The aforementioned inconsistencies in the methodologies of included studies produced quite heterogeneous results.

In a study carried out by Ghulam et al., more than one-third of the patients with stable AAA diameters appeared to have a growing AAA volume [9]. More patients from this group compared to those with a stable diameter and volume underwent aortic repair (20% vs. 5%). None of these patients developed a ruptured AAA, and most were treated electively. Only one patient had a symptomatic AAA. All decisions regarding aneurysm repair were made relying on diameter. The results of this study propose volume as a more sensitive tool for evaluating and following AAAs. On the other hand, it raises doubts about whether volume measurement would have any clinical relevance in daily practice. Khan et al. found that volume measurements correlate (r = 0.46) better than diameter with AAA growth [18]. It could be argued whether such a correlation is sufficient to rely completely on this parameter in clinical practice. A few studies included in this systematic review disclose the absence of diameter change with a significant change in volume without having a clear conclusion of such findings [23,28]. Authors note that volume measurement might add important information about AAA expansion and emphasize that more outcome-related studies are needed to assess the clinical significance of volume measurements.

Large-sample-size future studies comparing AAA diameter and volume impact on aneurysm prognosis, growth, and rupture rates are required. Using strictly standardized measurement criteria and having clearly defined expected outcomes could help gather higher quality data and assist in making unbiased decisions regarding the use of diameter and volume measurement of AAAs in daily clinical practice.

## 5. Conclusions

The clinical relevance of abdominal aortic aneurysm volume measurement is still unclear. Although studies show favorable and promising results for volumetric changes in AAA, especially in follow-up after EVAR, more standardized data are needed to further evaluate the importance of volumetric parameters in AAA. It remains unclear whether volume could replace or be a necessary addition to diameter measurement in assessing the risk of AAA rupture or during the follow-up after EVAR procedures.

## Figures and Tables

**Figure 1 biomedicines-11-00941-f001:**
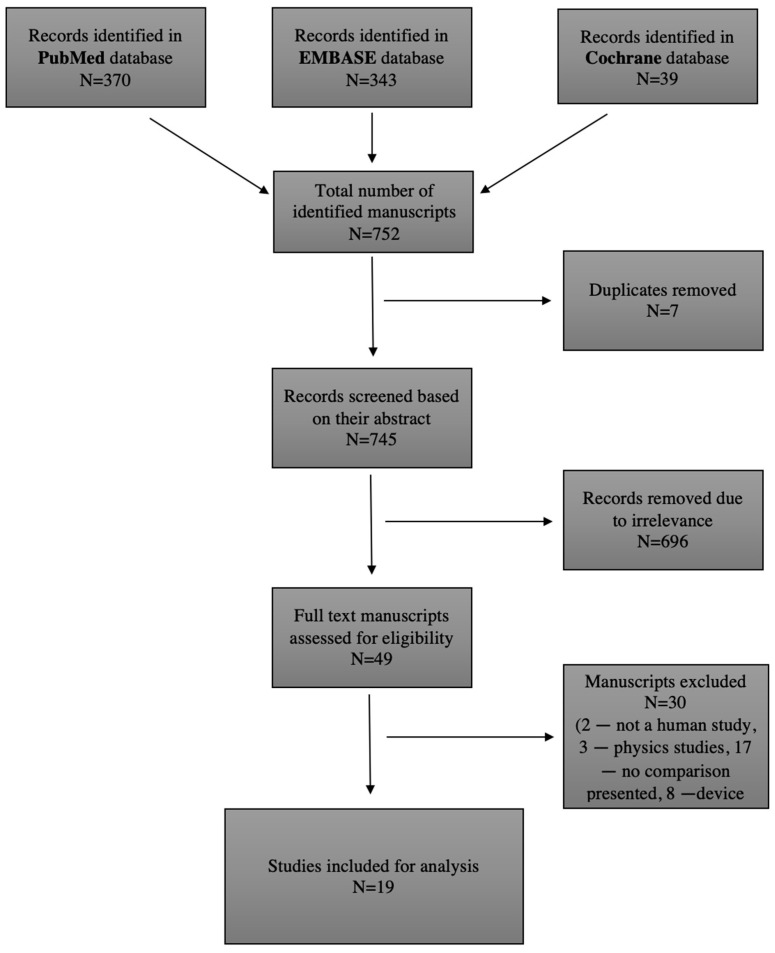
Literature search strategy and outcomes.

**Table 1 biomedicines-11-00941-t001:** Inclusion and exclusion criteria.

Inclusion Criteria	Exclusion Criteria
1. Publications regarding diameter and volume measurement in diagnosis and prognosis of AAA	1. Investigation of a different morphological criteria
2. Manuscripts in English, German, and Russian	2. Only abstract available
3. Human studies	3. Physiology reviews
4. Full text available	4. Studies with the physical background
	5. Case series and case reports

Abbreviations: AAA—abdominal aortic aneurysm.

**Table 2 biomedicines-11-00941-t002:** Quality assessment according to MINORS criteria.

Study	A Clearly Stated Aim	Inclusion of Consecutive Patients	Prospective Collection of Data	Endpoints Appropriate to the Aims of Study	Unbiased Assessment of the Study Endpoint	Follow-Up Period Appropriate to the Aim of the Study	Loss to Follow-Up Less Than 5%	Prospective Calculation of the Study Size	An Adequate Control Group	Contemporary Groups	Baseline Equivalence of Groups	Adequate Statistical Analyses	Total Max NON RCT 16, RCT 24
Wolf et al. (2002) [16]	2	2	1	2	2	1	1	0	n/a	n/a	n/a	n/a	11
Skrebunas et al. (2019) [17]	2	1	2	1	0	2	0	2	n/a	n/a	n/a	n/a	10
Ghulam et al. (2017) [9]	2	2	2	2	2	2	1	2	n/a	n/a	n/a	n/a	15
Khan et al. (2022) [18]	2	2	1	1	2	2	2	2	n/a	n/a	n/a	n/a	14
Quan et al. (2019) [19]	2	2	1	2	2	2	2	0	n/a	n/a	n/a	n/a	13
Bargellini et al. (2005) [20]	2	2	0	2	2	2	2	0	n/a	n/a	n/a	n/a	12
Fillinger et al. (2006) [21]	2	2	0	2	2	2	2	0	n/a	n/a	n/a	n/a	12
Wever et al. (2000) [22]	2	2	1	1	2	2	1	2	n/a	n/a	n/a	n/a	13
Parr et al. (2013) [23]	2	2	2	2	2	1	2	0	n/a	n/a	n/a	n/a	13
Schnitzbauer et al. (2018) [24]	2	2	0	2	2	1	2	0	n/a	n/a	n/a	n/a	11
Olson et al. (2022) [25]	1	1	2	1	0	2	2	2	n/a	n/a	n/a	n/a	11
Tzirakis et al. (2019) [26]	2	2	0	2	2	1	2	0	n/a	n/a	n/a	n/a	11
Kontopodis et al. (2014) [5]	2	2	0	2	2	1	2	0	n/a	n/a	n/a	n/a	11
Spanos et al. (2020) [6]	2	2	0	2	2	0	0	0	2	2	2	2	16
Liljeqvist et al. (2016) [27]	2	2	0	2	2	1	2	0	n/a	n/a	n/a	n/a	11
Renapurkar et al. (2012) [28]	2	1	0	2	2	1	2	0	n/a	n/a	n/a	n/a	10
Franchin et al. (2021) [29]	2	1	2	1	0	1	2	1	n/a	n/a	n/a	n/a	10
Kritpracha et al. (2004) [30]	2	0	0	2	2	1	1	0	n/a	n/a	n/a	n/a	8
Raghavan et al. (2000) [31]	2	0	0	2	2	0	2	1	1	2	0	2	14

Abbreviations: n/a—non applicable.

**Table 3 biomedicines-11-00941-t003:** Summary of studies that found no difference in diagnostic and prognostic values of volume and diameter measurement.

Study	Country	Size	Inclusion Criteria	Imaging Modality	Results	MINORS Criteria
Wolf et al. (2002) * [16]	USA	154	Patients after elective EVAR of AAA	CTA	The predictive values of changes in volume for identifying the presence or absence of endoleak were not significantly different from those associated with changes in transverse or orthogonal diameter.	11/16
Skrebunas et al. (2019) [17]	Lithuania	39	Patients before and after elective EVAR of AAA	CTA	Diameter increased in 11 (28.2%) of 39, but volume increased in 12 (30.8%). A moderate positive linear correlation between diameter and volume (R^2^ = 0.731, *p* < 0.0001). A clinically irrelevant AAA diameter increase after EVAR was observed in 8 (72.7%) of 11 cases. The AAA volume changes were also evaluated in those cases. There was no statistically significant difference between diameter and volumetric AAA changes in those cases (*p* = 0.184).	10/16

Abbreviations: MINORS—methodological index for non-randomized studies; EVAR—endovascular aneurysm repair; AAA—abdominal aortic aneurysm; CTA—computed tomography angiography.

**Table 4 biomedicines-11-00941-t004:** Summary of studies favoring volume measurement.

Study	Country	Size	Inclusion Criteria	Imaging Modality	Results	MINORS Criteria
Ghulam et al. (2017) [9]	Denmark	179	Patients with small (30–55 mm) AAAs	US and 3D US	Post-hoc analysis of the time period between the end of follow-up and manuscript preparation revealed that 14 patients underwent aortic repair: 13 elective repair (EVAR: *n* = 9; open repair: *n* = 4) and one subacute EVAR because of a symptomatic AAA. In this time period, more patients with a previously stable diameter and growing volume were growing in diameter, and more patients from this group than patients with a stable diameter and stable volume underwent aortic repair (20% vs. 5%).	15/16
Khan et al. (2022) [18]	UK	128	Patients with AAAs (30–70 mm in diameter)	3D US	AAA growth correlated more closely with AAA volume than diameter (r 0.46, *p* < 0.01). Aneurysm growth is most strongly related to AAA volume and inversely to wall volume, a more reliable way to measure wall thickness. A surveillance program that incorporates aneurysm volume and wall volume rather than just diameter may better inform surveillance intervals and surgical decisions.	14/16
Quan et al. (2019) * [19]	South Korea	82	Patients before and after EVAR of AAA	CTA and MRA	The enlargement rate of aortic volume was significantly different from the enlargement rate of Dmax (*p* = 0.02 by Wilcoxon rank-sum test). The occurrence of endoleaks between the Dmax-enlargement group and the no-enlargement group was significantly different (11, 100% vs. 19, 26.76%, *p* < 0.001). There was a significantly different rate of occurrence of endoleaks between the aortic volume enlargement group and the no-enlargement group (20, 90.91% vs. 10, 16.67%, *p* < 0.001). In the aortic volume enlargement group, there were more patients with endoleaks.	13/16
Bargellini et al. (2005) * [20]	Italy	63	Patients after EVAR of AAA	US, CTA	Endoleaks were found in 19 patients and were more frequent (*p* = 0.04) in patients with higher pre-procedural Dmax. The accuracy of volume changes in predicting endoleaks ranged between 74.6% and 84.1% and was higher than those of Dmax modifications. The strongest independent predictor of endoleak was a volume change at 6 months less than 0.3% (*p* = 0.005), although 6 of 19 (32%) patients with endoleak showed no significant AAA enlargement, whereas in 6 of 44 (14%) patients without endoleak the aneurysm enlarged.	12/16
Fillinger et al. (2006) [21]	USA	112	Patients with enlarging aneurysms (5-mm increase by Core laboratory or site) and at least 4 years of follow-up in the Excluder	CTA	A total of 38 AAAs (34%) were identified as enlarging. Of the 158 scans with a prior scan for comparison, 41% demonstrated growth relative to the initial scan by diameter criteria, but 79% demonstrated growth relative to the initial scan by 3-dimensional volume criteria. This difference was most evident at early time points: at 1 year, diameter criteria indicated that 8% of these AAAs were enlarging, but 56% were already enlarging by volume criteria. On average, enlargement was detected by volume 18 months before it was detected by diameter (18 vs. 36 months, *p* < 0.0001) and at a smaller diameter (55.4 mm vs. 59.8 mm; *p* < 0.0001).	12/16
Wever et al. (2000) * [22]	USA	35	Patients after EVAR of AAA	CTA	There was a poor correlation between the endoleak status and aneurysm growth, but the correlation between volume increase and endoleak was stronger (r = 0.37 at 6 months, r = 0.25 at 12 months) than the correlation between Dmax and endoleak (r = −0.07 and r = 0.11, respectively).	13/16
Parr et al. (2013) [23]	Australia	57	Patients with AAAs (25–55 mm in diameter)	CTA	A total of 42% of patients who had increased aortic volume above the upper 95% limit of agreement had no diameter change.	13/16
Schnitzbauer et al. (2018) * [24]	Germany	100	Patients after elective EVAR of AAA	CTA	The use of the reporting standard showed that the diameter measurements failed to detect aneurysm volume increase in 61–72% of cases with persistent type II endoleak.	11/16
Olson et al. (2022) [25]	USA	250	Patients with AAAs (35 mm to 50 mm male and 35–45 mm female)	CTA	The tortuosity index is associated with volume but not Dmax (difference 32.8 cm^3^/year, *p* < 0.001). Baseline volume accounted for more volume growth than Dmax (30% vs. 13%, *p* < 0.001). Predictors of volume growth: high baseline volume (regression coefficient 0.2, *p* < 0.001), tobacco use, tortuosity index (*p* < 0.001), and absence of diabetes. More tortuous aneurysms at baseline had significantly larger volume growth rates (difference, 32.8 cm^3^/year; *p* < 0.0001).	11/16
Tzirakis et al. (2019) [26]	Greece	30	Patients with AAAs	CTA	Statistical analysis showed strong evidence of a strong correlation between Dmax and volume growth rates (rho: 0.68, *p* < 0.001). In addition, there was strong evidence of a moderate correlation between Dmax growth and average surface growth (rho: 0.59, *p* < 0.001) and a moderate correlation with maximum surface growth (rho: 0.6, *p* < 0.001). Finally, there was strong evidence of a very strong association of volume growth with average surface growth (rho: 0.91, *p* < 0.001) and a strong association between volume growth and maximum surface growth (rho: 0.7, *p* < 0.001).	11/16
Kontopodis et al. (2014) [5]	Greece	34	Patients with AAAs (initial maximum diameter 40–53 mm)	CTA	There was a strong correlation between volume and Dmax growth rates (Spearman’s rho 0.6, *p* = 0.002). A total of 12 of the 15 AAAs having undergone surgical correction were in the high growth rate and only 3 in the low growth rate volume group (*p* = 0.005). With regard to the need for surgical repair, likelihood ratios between AAAs in the high and low growth rate groups were calculated (Likelihood ratio = 10) as well as sensitivity/specificity of median growth rates (Sensitivity = 80% Specificity = 74%).	11/16
Spanos et al. (2020) [6]	Greece	62	Patients with large ruptured (31) and unruptured (31) AAAs	CTA	The total aneurysm volumes for elective vs. rAAAs (*p* = 0.014) and true lumen volumes (*p* = 0.022) were significantly different between the groups. Maximum diameter did not have a statistically significant difference between the groups (*p* = 0.150). ROC curve showed that total aneurysm volume could predict rupture (AUC 0.68, *p* = 0.042). A threshold value of 380 mL was fairly well associated with rupture, with 60% sensitivity and specificity. Maximum diameter was not a predictor of rupture (AUC 0.62, *p* = 0.151).	16/24
Liljeqvist et al. (2016) [27]	Sweden	41	Patients with AAAs	CTA	Diameter correlated with volume with respect to baseline value (r = 0.71, *p* < 0.0001) and growth rate (r = 0.55, *p* = 0.0002). Significant absolute volume growth rate correlated stronger than significant absolute diameter growth rate with PWS (95% CI, 0.093–1.18) and PWRI (95% CI, 0.11–1.16) change rates.	11/16
Renapurkar et al. (2012) [28]	USA	100	Patients with AAAs	CTA	The correlation between diameter change and volume change was modest (r^2^ = 0.34; *p* = 0.001). Most patients (*n* = 64) had no measurable change in maximal diameter between studies (≤2 mm), but the change in volume was found to vary widely (−2 to 69 mL).	10/16
Franchin et al. (2021) * [29]	Italy/USA	149	Patients before and after EVAR of AAA	CTA	Diameter shrinkage was in 27 (18.1%), and volume shrinkage in 42 (28.2%). The presence of a persistent endoleak was associated with the absence of diameter shrinkage (*p* = 0.045; HR, 3.49; 95% CI, 1.031–11.859) and volume shrinkage (*p* = 0.001; HR, 7.75; 95% CI, 2.282–26.291). The ROC analysis demonstrated fair discrimination for this multivariate model (AUROC, 0.61; 95% CI, 0.52–0.71) with a 65.8% positive predictive value of no volume change in the presence of a persistent endoleak. Although the absolute diameter decrease had similar reliability to volume decrease, the volume analysis was more sensitive in assessing sac shrinkage.	10/16
Kritpracha et al. (2004) [30]	USA	68	Patients of after EVAR of AAA	CTA	The majority of increased AAA size studies were not detected by diameter measurement methods. Antero–posterior diameter was the least sensitive (15%) in detecting an increase in AAA size, particularly in the later postoperative period. Of the 55 studies with unchanged Dmax, 15 (27%) studies showed significant volume increase.	8/16
Raghavan et al. (2000) [31]	USA	7	6 AAA patients awaiting repair. One control subject without an aneurysm.	3D reconstructions of CTA	Of the factors studied (diameter, height, volume, systolic pressure), AAA volume appears to have the strongest correlation with PWS. Correlation coefficient 0.7.	14/24

Abbreviations: MINORS—methodological index for non-randomized studies; EVAR—endovascular aneurysm repair; AAA—abdominal aortic aneurysm; US—ultrasonography; 3D—three dimensional; CTA—computed tomography angiography; MRA—magnetic resonance angiography; Dmax—maximum diameter; rAAA—ruptured abdominal aortic aneurysm; ROC—receiver operating characteristic, AUC—area under curve; PWS—peak wall stress; PWRI—peak wall rupture index.

## Data Availability

The authors confirm that data supporting the findings of this study are available within the article.

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
