# Peer review of "Abdominal Aortic Aneurysm Diameter versus Volume: A Systematic Review"

_biomedicines, 2023, doi:10.3390/biomedicines11030941_

Round 1

Reviewer 1 Report

This manuscript is a systematic review of the results of previous studies comparing AAA diameter and volume measurements. In the result, the authors suggest that volume measurement is a potential approach to assess the risk of AAA rupture or during the follow-up after EVAR procedures. There are some weaknesses here.

Major

1.    the results section simply presented the results of the included articles in the Tables. The authors have done a lot of work, Why the authors did not further analyze the results of the extraction, such as forest figure.

2.     Different methods of measuring volume were used. Do the authors think this would affect the results?

Minor

1. Please state in the article the reasons for excluding articles written before the year 2000.

Author Response

Dear reviewer,

Thank you for your valuable comments. Please see the attachment.

Reviewer 2 Report

­The proposed review paper is devoted to analysis of the clinical relevance of abdominal aortic aneurysm volume measurement.

Preliminaries to the research area are provided. The importance of the aortic diameter for diagnosing and elective repair of abdominal aortic aneurysms is mentioned. The disadvantages of the method of diameter measurement are described. The recent alternative surveillance method of volume measurement is presented and studied in detail using recent scientific sources. Possible techniques for volume measurement are listed and various eventual advantages of this methodology are described.

The authors perform a detailed comparison between the methods of  diameter and volume measurements in the diagnosis and prognosis of abdominal aortic aneurysm studying a rich set of sources in English, German and Russian. The strategy and outcomes of the search of literature are presented. The selection criteria for the sources are formulated.    

The results of the comparison are described and analyzed in detail. Future research plans are presented.

The presentation of the main results is clear. From a formal point of view, all the contents seems to be correct. The results are valuable and worthy of being published taking into account their possible applications in clinical practice, in particular for diagnosis, surveillance, and clinical decision-making of abdominal aortic aneurysm. 

Minor revisions are suggested to improve the quality of the exposition:

p. 1, line 19: The meaning of the numer n=1650 is not clear. It is related with 752 manuscripts and 19 studies, but what is it’s meaning?

p. 1, lines 37-38: There is a repetition of the word “even”, I suggest eventual reformulation.

p. 4, lines 95, 97, 99: The formatting of the subsection’s titles need improvement. 

p. 4, line 120: The formatting of the Table 2 needs improvement.

Author Response

(The authors gave the same response as above.)

Reviewer 3 Report

The diameter of an aortic aneurysm is an important, but not the unique element of risk: you can discuss the value of intraluminal thrombus, walls thickness. diffuse calcifications. and the poorly known vasa vasorum pathology.  Today, the 3D imaging is of relevant importance, as well as 4D MRI angiography.  Therefore precedent data, without these parameters, have a less value. You have to consider this. Some references are badly cited. 

Author Response

(The authors gave the same response as above.)

Round 2

Reviewer 3 Report

The aneurysm diameter is the simplest index of impending rupture. You do not mention are characteristics, such as wall thickness and wall shear stress, that today can be assessed in 3D reformatted computed CT/MRI imaging.  The other hemodynamic features, such as aneurysm tortuosity, iliac arteries obstruction, etc. are more difficult to assess. Besides, inflammatory/ischemic-associated lesions could be detected by PET/CT scanning.  This invites us to consider a fundamental risk of rupture a progressive increase in diameter. You could discuss.

Author Response

Thank you for your comments.

The discussion has bean edited considering your comments. Lines 177-184.